# ORAG: Ontology-Guided Retrieval-Augmented Generation for Theme-Specific Entity Typing

**Jinfeng Xiao**[*] **& Linyi Ding**[*]
University of Illinois at Urbana-Champaign
Urbana, IL 61801, USA
{jxiao13,linyid2}@illinois.edu

**James Barry, Mohab Elkaref & Geeth De Mel**
IBM Research
Daresbury, England, UK
{james.barry,mohab.elkaref}@ibm.com
geeth.demel@uk.ibm.com

**Jiawei Han**
University of Illinois at Urbana-Champaign
Urbana, IL 61801, USA
hanj@illinois.edu

## Abstract

Large language models (LLMs) incorporated with retrieval-augmented generation (RAG) have shown great power in many NLP tasks, including fine-grained entity typing (FET). However, we observe that recent LLMs can easily suffer from hallucinations on highly specialized and fast-evolving themes (e.g., redox-active organic electrode materials), especially in the following cases: (1) *unseen entities*: an entity never appears in the pre-training corpora of LLMs; and (2) *misleading semantics*: the context of an entity can potentially mislead an entity typing algorithm if the relevant knowledge is not correctly retrieved and utilized. To address these challenges, this paper proposes an *Ontology-Guided Retrieval-Augmented Generation* (*ORAG*) approach that incorporates ontology structures with RAG for the theme-specific entity typing task. ORAG first enriches the label ontology with external knowledge and constructs a structured knowledge unit for each node. Then, it retrieves the relevant nodes by dense passage retrieval and expands the retrieved results based on the ontological structure. In this way, more supporting knowledge will be retrieved within the limited input of LLMs for entity typing. In the evaluation, we construct a dataset with two themes for theme-specific entity typing with a focus on unseen entities and misleading semantics. We observe notable cases of hallucination when vanilla RAG is applied to Llama-3, GPT-3.5, and GPT-4, while ORAG can effectively mitigate such hallucinations and improve the results.

## 1 Introduction

Retrieval-augmented generation (RAG) (Lewis et al., 2020) is an effective solution to deal with the limited input text length of LLMs when a task heavily relies on an external knowledge base like long documents or a large database, referred to as a retrieval memory. Given an LLM query, RAG first retrieves the relevant text chunks from the retrieval memory and then appends them to the query to generate answers. This RAG framework has shown its success on various NLP tasks.

Fine-grained entity typing (FET) is the task of classifying entities into detailed types, where RAG can take the large label ontology as the retrieval memory. Existing datasets of FET such as OntoNotes (Gillick et al., 2014) and FIGER (Ling & Weld, 2021) mainly focus on common, general topics in the real world such as *company*, *city*, and *actor*, which are easy to understand by LLMs since the context and entity types frequently appear in the pre-training corpora.

---

[*]Equal contribution

However, we observe that directly performing RAG with recent LLMs to zero-shot entity typing can easily lead to hallucinations on **highly specialized or fast-evolving themes**. This is observed with Llama-3 and GPT-4, which are among the most powerful open-source LLMs and commercial LLMs respectively.

We take a passage under a highly specialized theme "redox-active organic electrode materials" as example. The passage discusses the effectiveness of a novel electrode material Cu-TCA compared to a bare electrode. The FET task is to find the type of the "bare electrode" in a given fine-grained label ontology for the theme. The correct type is "working electrode".

> **Example Passage 1.** CV data of Cu-TCA were recorded using a Li foil as counter electrode. ... (Omitted) ... The CV curve of a **bare electrode** (only containing the Al foil) proves that the above-mentioned redox areas originate from the active sites of the Cu-TCA structure.

To reason the correct answer "working electrode", the model should (i) understand each type in the ontology to identify the correct type; and (2) understand the terms in the context such as *counter electrode*. RAG with GPT-4 gets the wrong answer "aluminum insertion electrode" because it semantically matches the misleading contextual information of the Al foil (which is an aluminum electrode) and Cu-TCA (which is an insertion electrode).

Inspired by such cases, we propose the task of **theme-specific entity typing**. Compared to FET, our task focuses on narrow themes, especially highly specialized or fast-evolving themes, which require expert knowledge to reason the entity types. In this task, the hallucinations of LLMs are more likely to happen in: (i) *unseen entities*: new entities that never appear in the pre-training corpora of LLMs (e.g., the entity in Figure 1); and (ii) *misleading semantics*: the context can potentially mislead an entity typing algorithm if the relevant knowledge is not correctly retrieved and utilized. Challenges in RAG include (i) lack of expert knowledge to understand the theme-specific label ontology; and (ii) even with a large collection of expert knowledge, the ineffective retrieval based on text chunks may miss evidence for reasoning with the limited input length of LLMs.

To tackle the challenges and mitigate LLMs' hallucinations for theme-specific entity typing, we propose the **Ontology-guided Retrieval-Augmented Generation (ORAG)**[1]. The overall design is outlined in Figure 2. For a better understanding of the types in the ontology, ORAG enriches the theme ontology with expert knowledge from multiple external sources. After the enrichment, each node in the theme ontology becomes a *structured knowledge unit* containing a knowledge-dense sub-unit and a semantic-rich sub-unit, which constitutes the *structured retrieval memory*. For **more knowledge-dense retrieval**, instead of dividing information into fixed-sized and incoherent text chunks, we retrieve the knowledge units with dense passage retrieval (DPR) (Karpukhin et al., 2020) and selectively add their sub-units into the LLM's query. For **more accurate reasoning**, we expand the evidence set of retrieved nodes with their neighbors on ontology. For example, we add the definitions of child nodes to discriminate between the more fine-grained entity types. Finally, ORAG combines the evidence and the context to predict the entity type. In the above Example Passage 1, ORAG successfully predicts the correct entity type "working electrode". A more in-depth explanation of ORAG's success in this case is provided in Section 5.2.

As no publicly available datasets focus on our settings, we construct a dataset of two themes: (i) a *battery theme* "redox-active organic electrode materials", and (ii) a *game theme* "online posts for Genshin Impact". The former represents *highly specialized themes*, which usually feature complicated ontologies and terminologies. The latter represents *fast-evolving themes*, where new entities and knowledge emerge so quickly (i.e., a significant update every six weeks) that the pre-training corpora of publicly available LLMs always lag behind the most recent version of the game.

We compare the entity typing performance of ORAG versus vanilla RAG when applied to current state-of-the-art LLMs on the two themes. We observe that ORAG consistently provides a higher boost to Llama-3, GPT-3.5, and GPT-4. The effectiveness of ORAG is especially prominent in cases of unseen entities and misleading semantics.

---

[1]Additional resources about this work are available at `https://github.com/JinfengXiao/ORAG`

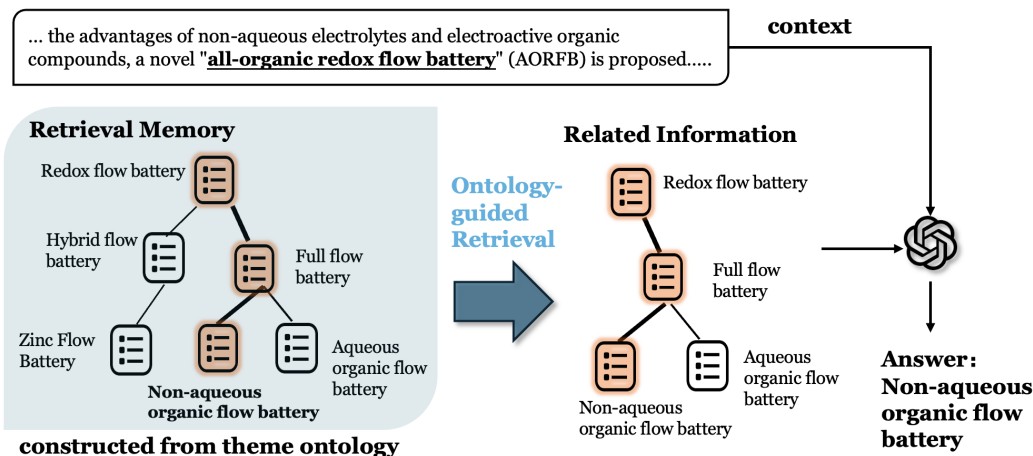

Figure 1: Ontology-guided RAG for theme-specific entity typing.

We summarize our contributions as follows:

- We propose the theme-specific entity typing task, discuss two types of hard cases for LLMs, and construct a dataset of two themes focusing on such hard cases.

- We propose ORAG, which performs structured retrieval on knowledge-enriched ontologies, to achieve more knowledge-dense retrieval and more accurate reasoning.

- We experimentally demonstrate that applying ORAG to the theme-specific entity typing task can generate more accurate results and mitigate the hallucinations of LLMs compared to vanilla RAG, without requiring additional labeling efforts.

## 2 Related Work

### 2.1 Fine-Grained Entity Typing

Compared to conventional entity typing, the label space of FET usually entails dozens or even hundreds of entity types, which is organized into a hierarchical ontology structure. Thus, FET faces the challenges of the large label space and limited accurate annotations. To deal with the data scarcity issue, ontologies and external knowledge are exploited and utilized in previous works. AFET (Ren et al., 2016) leverages the defined type hierarchy to formulate loss functions and integrates them into a unified optimization problem. ALIG-NIE (Huang et al., 2022) learns to relate entity types with vocabulary using the ontology and then uses a prompt-based method to enrich the few-shot training samples. SEType (Zhang et al., 2024) finds more instances for each seen type from an unlabeled corpus using pre-trained language models. OnEFET (Ouyang et al., 2023) enriches the original ontology structure with instances and topics. These methods are in need of annotated instances and thus not developed under a zero-shot setting.

For the zero-shot setting, a line of work utilizes linking from Wikipedia entries to the entities or types (Choi et al., 2018). For example, ZOE (Zhou et al., 2018) uses the taxonomy of Freebase and types the entity by linking it to Wikipedia entries. The assumption that such links exist is a limitation of this line of work. Several LLM-based methods are also proposed for zero-shot FET. OntoType (Komarlu et al., 2023) proposes an ontology-guided framework that leverages pre-trained language models and head words to match the fine-grained types to a type ontology. Qin et al. (2023) and Hu et al. (2024) apply LLMs to zero-shot entity typing by prompt engineering. These studies focus on common entity types, and the assumption that LLMs know the types well limits their applicability to highly specialized or rapidly evolving themes.

## 2.2 Knowledge-Grounded Retrieval-Augmented Generation

RAG (Lewis et al., 2020) retrieves text chunks from text corpora to augment LLM queries. This technique has shown its power on various NLP tasks (Gao et al., 2023). Besides unstructured text, structured knowledge like knowledge graphs (KGs) and ontologies can also be used as retrieval memory in the RAG framework. KG-RAG (Soman et al., 2023) applies RAG to biomedical KGs to facilitate domain-specific generation tasks. SURGE (Kang et al., 2023) trains graph neural networks to retrieve knowledge triples from a KG, which encourages the model to generate knowledge-grounded answers. KGLM (Logan et al., 2019) selects relevant facts from a KG with a KG embedding in each step of generation. KGQA (Sen et al., 2023) retrieves knowledge triples from a KG related to query entities by Hadamard product to augment the LLM input. REALM (Guu et al., 2020) retrieves documents from Wikipedia during the pretraining stage. EMAT (Wu et al., 2022) encodes external knowledge into a key-value memory to be retrieved and introduces a memory-augmented transformer for QA tasks. While these works attempt to retrieve evidence from KGs or ontologies, most of them ignore or do not fully utilize structured information. How to better utilize structured and unstructured knowledge for FET remains less studied.

# 3 Method

## 3.1 Problem Formulation

The conventional FET task mainly focuses on frequent and common entity types in the real world. Examples include *company*, *city*, and *doctor* in the FIGER dataset (Ling & Weld, 2021), as well as *date*, *money*, and *time* in the OntoNotes dataset (Gillick et al., 2014). Despite the large label space, the types are easy to understand for LLMs without requiring much additional expert knowledge (Ding et al., 2022). In contrast, our targeted theme-specific entity typing task involves several less-studied challenges. (1) It focuses on narrow themes that need in-depth exploration for specific purposes. (2) It often deals with entities, types, or contexts that are unseen or long-tailed in the pre-training corpora of publicly available pre-trained LLMs. (3) It requires expert knowledge and reasoning on the themes to figure out the answers.

For each theme, we assume its theme ontology (i.e., a label hierarchy) is available. As reviewed in Section 2.1, the availability of such an ontology is part of the problem definition of works on FET. Our task is to type the entities with the most proper fine-grained labels. Once a fine-grained type is identified, all parent types on the path from the root of the ontology to the fine-grained type are also considered as identified. Therefore, our task belongs to the family of hierarchical multi-label classification. Take the case in Figure 1 as an example. The theme ontology contains entity types like *non-aqueous organic flow battery*, which is the most proper fine-grained type of the underlined entity in the given context. Once this type is identified, all highlighted types in the ontology are considered as identified.

## 3.2 Structured Retrieval Memory Construction

### 3.2.1 Theme-Specific Ontology Enrichment

For a highly-specialized theme, such as *redox-active organic electrode materials*, the raw label ontology may not provide sufficient information for reasoning the entity types by LLMs. Directly applying LLMs or RAG on the label hierarchy can easily lead to hallucination, especially for knowledge-intensive contexts and unseen entities since the reasoning would require specialized expert knowledge.

To address the issue related to the lack of specialized information, we first enrich the label ontology with expert knowledge from external sources (Panel 1, Figure 2). For each entity type in the label ontology, we collect the following knowledge from various sources:

  i. The concise and formal **definition** from theme-specific dictionaries or encyclopedias can help establish a common understanding of highly specialized terms and avoid ambiguity.

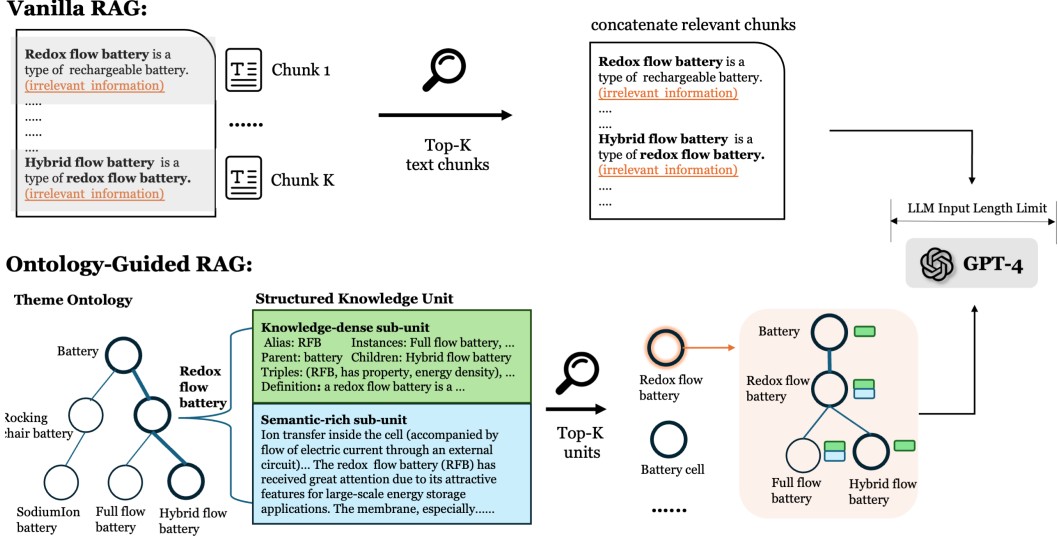

Figure 2: **Overall framework (ORAG vs. RAG)**. Conventional RAG retrieves text chunks while ORAG retrieves informative structured knowledge units. ORAG first enriches the ontology with external sources and constructs a knowledge unit for each node. It then retrieves the relevant nodes and further expands evidence based on the ontology structure.

ii. **Alias** like abbreviations can help to recognize entities that may appear under different names and improve the recall rate of retrieval.

iii. **KG triples** can provide possibly related entities that may be included in the context. The KGs usually contain many noisy relations. We filter the noisy relations with the similarity between the triples and the theme, calculated by a pre-trained encoder.[2]

iv. If some **entity instances** are available, we also collect them to help LLMs better understand long-tailed types. If rich information is available about such instances, we further add them into the ontology as leaf nodes under the corresponding type.

v. To include comprehensive knowledge for better reasoning, we collect related **descriptions** from large corpora such as Wikipedia, which can provide multi-view information about the type. A description is usually longer than a definition.

Note that we do not assume complete knowledge is available. In other words, we allow substantial misses in the above fields in most types, which is true in real-world applications. For example, in Figure 2, the real available data miss some information for all types except the *battery* type and the *redox flow battery* type (e.g., the *rocking chair battery* type misses knowledge triples, the *full flow battery* type misses everything except a definition). Our algorithm handles partial information by inheriting information between parents and children in the ontology, as described in later sections.

### 3.2.2 Structured Knowledge Unit Construction

As shown in Figure 2, vanilla RAG retrieves relevant text from its retrieval memory, which are directly split into fixed-size text chunks. However, the evidence for reasoning may be scattered in different chunks, and thus the retrieved chunks could include a considerable percentage of irrelevant context. With the limited input length of LLMs, including irrelevant context in one chunk means dropping other chunks that also contain some relevant information. Therefore, this type of naive retrieval can be inefficient in terms of the density and coverage of the relevant knowledge in the retrieved results. To effectively retrieve the

---

[2]https://huggingface.co/sentence-transformers/all-MiniLM-L6-v2

related information, we construct a **Structured Knowledge Unit** for each entity type in the theme ontology (Panel 1, Figure 2).

The structured knowledge unit of an entity type includes two parts:

i. **Knowledge-dense sub-unit**: It includes the concise definition, alias, KG triples, and instances.

ii. **Semantic-rich sub-unit**: It includes the descriptions about the entity type.

We separate the two sub-units to balance knowledge density and semantic richness when we augment LLM queries in later steps. The knowledge-dense sub-units contain the most essential and concise information about the entity types. Although the semantic-rich sub-unit may be generally less important and informative, it may also provide subtle signals for reasoning if it is aligned with an LLM query. Thus, the handling of the two sub-units is different.

We also observe that RAG has better performance on understanding and reasoning when the retrieval memory consists of complete sentences instead of key-value pairs. Therefore, we design patterns for each type of information. For example, for the alias of *redox flow battery*, we store the sentence *Redox flow battery is also known as RFB* in the retrieval memory instead of *(alias, RFB)*. Similarly, we restate other information with complete sentences and finally construct the retrieval memory.

### 3.3 Ontology-Guided RAG

#### 3.3.1 Evidence Retrieval Based on Structured Units

Different from vanilla RAG that searches text chunks, ORAG searches structured knowledge units. The motivation of ontology-guided RAG can be two-fold: (1) Structured knowledge units are more knowledge-dense and informative when compared to fixed-sized text chunks. With knowledge-dense units, ORAG can append more relevant evidence to query LLMs within the same input length to facilitate entity typing. (2) The retrieved information is self-coherent and effective and could contain less irrelevant information (which may mislead the LLMs to generate erroneous results (Wang et al., 2023)).

We use the embedding model [3] from OpenAI to encode the sub-units and cluster the embeddings based on FAISS index (Johnson et al., 2019). For dense passage retrieval (DPR) (Karpukhin et al., 2020), given the LLM input query for entity typing, we rank the structured knowledge sub-units in a bi-encoder architecture.

We retrieve the top-k entity types on the theme ontology regarding the rank of similarities of sub-units. If at least one sub-unit is retrieved, we consider the corresponding entity type to be a relevant node in the ontology. The entity types will be used as starting entities for further ontology-guided evidence expansion and selection.

#### 3.3.2 Ontology-Guided Evidence Expansion

In this section, we further expand and select the information based on ontology and starting entities (i.e. *redox flow battery* and *battery cell* in Figure 2).

The starting entity types are retrieved because they can be directly semantically matched by entity mentions or context. However, the one-hop semantically matching evidence may still be insufficient when the typing process requires reasoning, especially for unseen entities.

Take the example in Figure 1, the entity mention *all-organic redox flow battery* is a newly proposed entity unseen for LLMs. The mention is semantically matched to the type *redox flow battery*. However, it can be reasoned from the context that it belongs to a more fine-grained type *non-aqueous organic flow battery*. In the vanilla RAG, the descriptions of the target type are not retrieved, and LLMs generate an inaccurate answer.

---

[3]https://platform.openai.com/docs/guides/embeddings

Table 1: Dataset statistics. Battery represents "redox-active organic electrode materials" and Game represents "online game posts for Genshin Impact".

| Theme | Documents | Entities | Hard Cases | Types |
|---|---|---|---|---|
| Battery | 20 | 100 | 20 | 484 |
| Game | 20 | 200 | 20 | 73 |

The goal of this step is to include multi-hop information for more accurate reasoning on entity types. Given a starting entity, to discriminate the finer-grained entity type, we retrieve its top-N relevant child nodes. In the case above, the LLMs can give the right answer with the knowledge units of *non-aqueous organic flow battery* and *aqueous organic flow battery*. We also retrieve the knowledge-intensive sub-unit of its parent node to provide more information to discriminate the starting type and its sibling types. In this way, even if some knowledge about a type is missing, the information inherited from parent and child nodes can help.

Due to the limited input length of LLMs, for most retrieved entity types, we only include the concise knowledge-dense sub-units in the augmented LLM query. The long semantic-rich sub-units are included only when their similarity to context is higher than the knowledge-dense sub-unit, which means the descriptions can help understand the context. The evidence is gathered, and identical items are merged. Finally, the evidence and the context are fed into LLMs to perform typing. The query prompts are as follows, where ⟨entity⟩ and ⟨passage⟩ are placeholders for the corresponding entity and passage.

> **LLM query template.** What are the types of the entity ⟨entity⟩ in this passage? If there are multiple types, return them all. Format the output as type1, type2, … ⟨passage⟩

## 4 Experiments

### 4.1 Dataset Construction

We construct a dataset with two themes: one for *redox-active organic electrode materials* and the other for *online game posts for Genshin Impact*. The dataset have a focus on unseen entities and misleading semantics, two situations where LLMs are more likely to suffer from hallucinations. The former features highly specialized and complicated terminologies and ontology structures, whereas the latter features fast-evolving entities and knowledge that pre-training corpora of commercial LLM services almost always lag behind. The data on each theme include the following. (1) Corpus: a set of relevant documents that contain unseen entities and/or misleading semantics; (2) Ontology: a set of entity types arranged into a hierarchy describing the fine and coarse granularities; (3) Entity mentions: entity spans are located in the corpus by human annotators; (4) Entity types: the ground-truth entity types of the entity spans are identified from the given ontology by human annotators; and (5) Hard cases: A hard case is manually selected from each document, representing an entity mention that can hardly be typed correctly with its context only but correct typing becomes logically possible when theme-specific knowledge is included in the reasoning. In our experiments, such a hard case always features either unseen entities or misleading semantics. The statistics of the dataset are given in Table 1.

#### 4.1.1 Highly Specialized Theme: Redox-Active Organic Electrode Materials

The first theme is a highly specialized battery theme of *redox-active organic electrode materials*. We use the Battery Interface Ontology (BattINFO) (Clark et al., 2022) which contains around 20,000 entity types. Entity annotations are performed on a set of research papers in this field with a focus on batteries and electrodes. All annotated entities have types under the two sub-ontologies rooted at the Battery type (with 298 nodes in the sub-ontology) and the Electrode type (with 186 nodes in the sub-ontology). These two sub-ontologies add up to 484 types in total. The structured retrieval memory is constructed from BattINFO,

Wikipedia, Wikidata, and Electropedia[4] which contains all the terms and definitions in the International Electrotechnical Vocabulary.

### 4.1.2 Fast-Evolving Theme: Online Game Posts for Genshin Impact

The second theme is a fast-evolving game theme of *online posts for Genshin Impact*[5], a popular game winning the TGA best mobile game in 2021 and keeping fast-evolving on new entities. We collect bilingual passages from Chinese[6] and English[7] forums. We invited senior players with at least three years of gaming experience on Genshin Impact to construct the ontology (with 73 nodes) and perform annotation. The structured retrieval memory is constructed from Genshin Impact Wiki[8].

### 4.2 Experimented Methods

We report the results of the following methods: (1) **RAG-Llama-3**: the open-source Llama-3-8B-Instruct model[9] combined with vanilla RAG, (2) **RAG-GPT-3.5**: the GPT-3.5 with vanilla RAG, (3) **RAG-GPT-4**: the GPT-4 with vanilla RAG, (4) **ORAG-Llama-3**: the same Llama-3 with ORAG, (5) **ORAG-GPT-3.5**: the same GPT-3.5 with ORAG, and (6) **ORAG-GPT-4**: the same GPT-4 with ORAG. The pre-training data cutoff dates of both GPT-3.5 and GPT-4 are September 2021. All methods share the same prompts in Section 3.3.2.

### 4.3 Experiment Results

Table 2 shows the entity typing results on the two themes. We report the macro-average of precision, recall, and F1 scores. Given an LLM (Llama-3, GPT-3.5, or GPT-4), our ORAG improves the F1 scores over the vanilla RAG by 10% ∼ 24%. Remarkably, ORAG improves the precision of GPT-3.5 by 24% and 29% respectively, demonstrating the substantial efficacy of ORAG in reducing LLM hallucinations. ORAG also boosts the precision of GPT-4 to a nearly perfect level, providing hope for the complete elimination of hallucinations when the correct knowledge is used together with more advanced LLMs in the future. ORAG also boosts the performance of Llama-3 compared to vanilla RAG, demonstrating the applicability of our method to open-source LLMs.

A notable improvement in recall is also consistently observed, while this improvement has a different indication than that in precision. Without ORAG, LLMs tend to predict coarse-grained types, which is easier to get right. Stopping at correct but coarse-grained types without going deeper into the ontology only hurts the recall but not the precision. That is why the precision scores are consistently higher than recalls. The rise of recall brought by ORAG demonstrates the usefulness of theme-specific knowledge in guiding LLMs into more fine-grained types.

We also examined the performance of all models on the set of hard cases. Results are shown in Table 3. While vanilla RAG performs much worse on hard cases compared to easy cases, ORAG brings the typing quality on hard cases much closer to the overall performance. In the battery theme, the performance of LLMs on hard cases is almost brought back to that on the average cases, indicating that ORAG can effectively handle unseen cases and misleading semantics. On the game theme, the performance of GPT-3.5 on hard cases after ORAG is applied still has a gap from its counterpart on the average cases. As this theme is evolving so fast (i.e., a significant update every six weeks) that a considerable portion of the up-to-date knowledge is missing from LLM pre-training corpora, the strong reasoning capability of very large models seems to play an essential role in handling hard cases.

---

[4]https://www.electropedia.org/

[5]https://en.wikipedia.org/wiki/Genshin_Impact

[6]https://ngabbs.com/

[7]https://www.hoyolab.com/home

[8]https://genshin-impact.fandom.com/wiki/Genshin_Impact_Wiki

[9]https://huggingface.co/meta-llama/Meta-Llama-3-8B-Instruct

Table 2: Theme-specific entity typing results for **all entities**. Themes are abbreviated. Battery: Redox-active organic electrode materials. Game: Online game posts for Genshin Impact.

| LLM | Method | Battery Theme | | | Game Theme | | |
|---|---|---|---|---|---|---|---|
| | | Precision | Recall | Micro-F1 | Precision | Recall | Micro-F1 |
| LLAMA-3 | RAG | 0.86 | 0.56 | 0.68 | 0.76 | 0.61 | 0.67 |
| | ORAG | **0.97** | **0.80** | **0.83** | **0.85** | **0.80** | **0.83** |
| GPT-3.5 | RAG | 0.72 | 0.64 | 0.64 | 0.68 | 0.59 | 0.61 |
| | ORAG | **0.96** | **0.79** | **0.83** | **0.97** | **0.73** | **0.80** |
| GPT-4 | RAG | 0.87 | 0.65 | 0.70 | 0.92 | 0.71 | 0.78 |
| | ORAG | **0.99** | **0.90** | **0.94** | **0.98** | **0.82** | **0.88** |

Table 3: Theme-specific entity typing results for **hard cases** with unseen entities or misleading semantics. Themes are abbreviated. Battery: Redox-active organic electrode materials. Game: Online game posts for Genshin Impact.

| LLM | Method | Battery Theme | | | Game Theme | | |
|---|---|---|---|---|---|---|---|
| | | Precision | Recall | Micro-F1 | Precision | Recall | Micro-F1 |
| LLAMA-3 | RAG | 0.74 | 0.48 | 0.53 | 0.60 | 0.56 | 0.51 |
| | ORAG | **0.95** | **0.76** | **0.83** | **0.87** | **0.82** | **0.83** |
| GPT-3.5 | RAG | 0.67 | 0.43 | 0.50 | 0.42 | 0.41 | 0.39 |
| | ORAG | **0.99** | **0.71** | **0.79** | **0.94** | **0.50** | **0.62** |
| GPT-4 | RAG | 0.77 | 0.41 | 0.48 | 0.83 | 0.53 | 0.64 |
| | ORAG | **0.99** | **0.87** | **0.92** | **0.96** | **0.78** | **0.85** |

## 5 Case Studies

### 5.1 A Case of An Unseen Entity

In the following, we discuss a case of an unseen entity from the game theme. In Example Passage 2, the ground-truth type of the entity "fruit platter" is *Weapon → Catalyst* (from coarse-grained type to fine-grained type). This weapon was first online in February 2022, so the GPT models with a training corpora cut-off date in September 2021 never saw this entity before. The entity "fruit platter" is a nickname of the weapon Kagura's Verity originating from the community, and is also not included in databases. Therefore, the correct entity type should be logically inferred from theme-specific knowledge and the context.

In this case, the context discusses some characters and the weapons they can use. It can be semantically inferred from the context that the "fruit platter" refers to a *Weapon*. However, with expert knowledge about characters, we can further infer that "fruit platter" is *Catalyst*.

> **Example Passage 2.** (Original post in Chinese) ...冬极来了，公子可以用上自己的武器，宵宫也能拿回飞雷了。但是，在没有八重草神的情况下，万一歪了果盘，可以给谁用？...
> (Translated post in English) Now that the Polar is coming, Childe will have his own weapon, and Yoimiya can take back her Thunder. Unfortunately, I don't have Miko or Nahida. If I turn out to get the **fruit platter** instead, who can make use of it?

RAG-GPT-4 only gives the coarse-grained entity type *Weapon*. The *Weapon* is generated because the term can be directly semantically matched in the context. Also, the RAG-GPT-4 lacks essential knowledge about context to infer the more fine-grained entity type.

ORAG-GPT-4 correctly predicts the fine-grained type *Catalyst*. Although the *Catalyst* does not appear in the context, we still get its structured knowledge units by evidence expansion from its parent node *Weapon*. Also, ORAG retrieves the related information of two characters

in the context, Miko Yae and Nahida, which reveals that both Miko and Nahida use *Catalysts*. With such information added to the query, GPT-4 becomes able to tell the correct type.

## 5.2 A Case of Misleading Semantics

> **Example Passage 1 (Repeated).** CV data of Cu-TCA were recorded using a Li foil as counter electrode. . . . (Omitted) . . . The CV curve of a **bare electrode** (only containing the Al foil) proves that the above-mentioned redox areas originate from the active sites of the Cu-TCA structure.

In this section, we dive deeper into the "bare electrode" case in Example Passage 1 in Section 1 to analyze the output of experimented methods. We repeat the passage here for readers' convenience. The "bare electrode" has two ground-truth fine-grained types following the ontological paths: *Electrode → Working Electrode*, and *Electrode → Active Electrode → Metal Electrode → Aluminum Electrode*. (The *Aluminum Electrode* type is not discussed in Section 1 to avoid over-complication.) In this case, the correct typing requires a full understanding of the context and the types to avoid being misled by irrelevant context.

RAG-GPT-4 predicts Aluminium Electrode and Aluminum Insertion Electrode, where the former is a correct fine-grained type but the latter is a hallucination. While an Insertion Electrode type can be inferred from the first half of Example Passage 1 and an Aluminum Electrode type can be inferred from the second half, combining the two pieces of semantics into an Aluminum Insertion Electrode type is scientifically wrong.

ORAG-GPT-4 predicts Aluminium Electrode and Working Electrode, which are correct and fine-grained. The algorithm retrieves the structured knowledge unit of the Working Electrode through its evidence expansion process, as the Working Electrode is a child type of Electrode. It also retrieves the unit of Insertion Electrode through its parent Active Electrode. With the relevant knowledge added to the query, GPT-4 becomes able to identify that the Working Electrode is a match while the Insertion Electrode is not.

## 6 Conclusion

In this work, we introduce ORAG, a theme-specific fine-grained entity typing framework that works well on cases of unseen entities and misleading semantics that are hard for LLMs. Compared to vanilla RAG, our proposed ORAG gives more precise entity types with structured retrieval from knowledge-enriched theme-specific ontology. During the evaluation of a highly specialized theme on electrode materials and a rapidly evolving theme on game posts, we find that our ORAG yields more accurate theme-specific entity typing results compared to the vanilla RAG method.

## Acknowledgement

The research was supported in part by the IBM-Illinois Discovery Accelerator Institute (IIDAI), the Molecule Maker Lab Institute: An AI Research Institutes program supported by NSF under Award No. 2019897, INCAS Program No. HR001121C0165, National Science Foundation IIS-19-56151, and the Institute for Geospatial Understanding through an Integrative Discovery Environment (I-GUIDE) by NSF under Award No. 2118329. Any opinions, findings, and conclusions or recommendations expressed herein are those of the authors and do not necessarily represent the views, either expressed or implied, of IBM or the U.S. Government.

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
