# OpenReview forum: "ORAG: Ontology-Guided Retrieval-Augmented Generation for Theme-Specific Entity Typing"
_colmweb.org/COLM/2024/Conference — COLM_

### Official Review · Reviewer_pCEi · 2024-05-10

**Rating:** 7
**Confidence:** 4
**Ethics Flag:** 1

**Summary:**

- This paper focuses on theme-specific entity typing. Compared to typical fine-grained entity typing, theme-specific entity typing requires expert knowledge to reason the entity types, which causes LLMs to be in trouble of hallucinations in two cases. First, unseen entities: an entity never appears in the pre-training corpora of LLMs. Second, misleading semantics: the context of an entity can potentially mislead an entity typing algorithm if the relevant knowledge is not correctly retrieved and utilized.
- This paper clearly describes two technical challenges to address: (i) lack of expert knowledge to understand the theme ontology; and (ii) even with a large collection of expert knowledge, the ineffective retrieval based on text chunks may miss evidence for reasoning with the limited input length of LLMs.

**Reasons To Accept:**

- Overall, the proposed framework achieves remarkable performance improvement.
- Also, the proposed framework successfully deals with the problematic cases: (i) unseen entities and (ii) misleading semantics.

**Reasons To Reject:**

- The proposed framework consists of two main techniques: (i) Structured Retrieval Memory Construction (Section 3.2) and (ii) Ontology-Guided RAG (Section 3.3). Unfortunately, this paper does not investigate the effectiveness of each technique. Not a few readers would like to know to what extent each technique contributes to performance improvement.

---

> ### Author Rebuttal · Authors · 2024-05-31
>
> Thank you for your positive feedback, especially on our technical contributions and experimental success.
>
> For your concern about the **effectiveness of each technique**, we provide **ablation study** results on the battery theme in the following tables. The experiment analyzes the effectiveness of (i) the text unit, (ii) the knowledge unit, and (iii) the evidence expansion process.
>
> | Model          | Precision | Recall | F1   |
> |----------------|-----------|--------|------|
> | RAG-GPT-3.5    | 0.72      | 0.64   | 0.64 |
> | ORAG-GPT-3.5 (No text unit) | 0.79 | 0.78 | 0.78 |
> | ORAG-GPT-3.5 (No knowledge unit) | 0.75 | 0.58 | 0.53 |
> | ORAG-GPT-3.5 (No expansion) | 0.85 | 0.70 | 0.74 |
> | **ORAG-GPT-3.5** | **0.96** | **0.79** | **0.83** |
>
> | Model          | Precision | Recall | F1   |
> |----------------|-----------|--------|------|
> | RAG-GPT-4      | 0.89      | 0.64   | 0.75 |
> | ORAG-GPT-4 (No text unit) | 0.83 | 0.81 | 0.78 |
> | ORAG-GPT-4 (No knowledge unit) | 0.85 | 0.77 | 0.74 |
> | ORAG-GPT-4 (No expansion) | 0.87 | 0.85 | 0.81 |
> | **ORAG-GPT-4** | **0.99** | **0.90** | **0.94** |
>
> We welcome further comments and discussions.

---

> > ### Comment · Reviewer_pCEi · 2024-06-05
> >
> > Thank you for the response.
> > I think that the ablation study makes clearer how much each technique contributes to the final results.
> > Thanks again.

---

> > > ### Author Response · Authors · 2024-06-06
> > >
> > > Thank you for your positive and constructive feedback!

---

### Official Review · Reviewer_YRhV · 2024-05-11

**Rating:** 7
**Confidence:** 4
**Ethics Flag:** 1

**Summary:**

The paper proposes a new task called theme-specific entity typing which is related to the typical entity typing task problems studied in NL, but for a specific theme or domain such as online gaming. The paper aims to guide the LLM generation with information-rich ontologies. It organizes the knowledge in a hierarchical ontology with each node representing an ontology type along with knowledge-dense sub-units and semantic-rich sub-units. The retrieval happens at the ontology’s node level as opposed to the chunk of text used in regular RAG.  The paper curated two datasets with human annotations and the corresponding ontologies. Experimental results show that the ontology-guided RAG makes GPT-3.5 and 4 models perform better.

**Reasons To Accept:**

1. The problem statement is unique. It will be useful for entity typing in highly specialized domains where information-rich domain-specific ontology is available.
2. The paper shows the effectiveness of ontology-guided RAG as opposed to traditional RAG with strong results.
3. The paper contributes 2 datasets which (once made public) will be useful for theme-specific entity typing research in future research.

**Reasons To Reject:**

1. The  experiment setup does not quite reflect the problem statement. The problem statement aims to solve fine-grained entity typing under “”highly-specialized and fast-evolving themes” and yet, the experiment setup requires an information-rich ontology with type description from specific dictionaries and encyclopedias, aliases, KG triples, etc. as outlined in section 3.2.1. How is it possible to have this complete set of knowledge for a “fast-evolving theme”? It seems the paper makes a strong assumption that such information is readily available for emerging entity and entity types. In practice, it will be hard to satisfy this assumption.
2. The experiment section is not extensive.
- I understand that it is time consuming to curate annotated data, I’d still expect evaluation to be done on more datasets.
- Lack of experiments with open-sourced LLMs. The paper only used GPT-3.5 and GPT-4 for experiments which will be hard to reproduce. It’s worth exploring similar techniques for other open-sourced LLM so that experimental results can be reproduced.
- Lack of ablation studies. For example, what happens if the “Semantic-rich sub-unit” is removed from the ontology, or the descriptions and the aliases are removed? Having an ablation study would help to better understand the contribution of each piece of information in RAG.

*After rebuttal*
The authors have addressed my questions. I have updated my scores accordingly.

---

> ### Author Rebuttal · Authors · 2024-05-31
>
> Thank you for recognizing our work’s uniqueness, effectiveness, and usefulness. We aim to respond concisely within the length limit.
>
> **The Availability of Knowledge**
>
> Thank you for raising the question on the availability of knowledge for fast-evolving themes. We would like to clarify the following points.
>
> - **We do not require a complete set of knowledge.**  In fact, most types in our ontology are described by partial information, and our reported results are based on such partial information. In Figure 2, all types miss some information except the Battery type and the Redox Flow Battery type (e.g., the Rocking Chair Battery type misses knowledge triples, the Full Flow Battery type misses everything except a definition)
> - **Our algorithm handles partial information by inheriting information between parents and children in the ontology**. For example, although Full Flow Battery misses everything except a definition, by inheriting the information from its parent type, Redox Flow Battery, LLMs become better at tackling the Full Flow Battery type.
> - **We assume partial knowledge is available. This is true even for fast-evolving themes.** On one hand, entity types evolve much slower than entities, and information for existing entity types is largely available for fast-evolving themes. On the other hand, new knowledge regarding emerging types may be extracted from texts (e.g., [Klink-2](https://skm.kmi.open.ac.uk/klink-2/)), which is out of the scope of this study, or updated by the users to customize the results.
>
> **The Scope of Experiments**
>
> Our response to Reviewer 28vN is dedicated to this concern. To save space, please refer to that response. There, we provided **new results on an additional theme** and reached the same conclusions.
>
> **Open-Source LLMs**
>
> Following your suggestion, we added an experiment with **LLAMA3**-8b-instruct, which is open-source and easy to reproduce. The **results are consistent** with the LLMs reported in our submission.
>
> | Battery Theme | Precision | Recall | F1 |
> | --- | --- | --- | --- |
> | RAG | 0.86 | 0.56 | 0.68 |
> | ORAG (Ours) | **0.97** | **0.80** | **0.87** |
>
> | Game Theme | Precision | Recall | F1 |
> | --- | --- | --- | --- |
> | RAG | 0.76 | 0.61 | 0.67 |
> | ORAG (Ours) | **0.85** | **0.80** | **0.83** |
>
> **Ablation Study**
>
> To save space, we provide the **results of an ablation study** in our response to Reviewer pCEi.
>
> We welcome further comments and discussions.

---

> > ### Comment · Reviewer_YRhV · 2024-06-06
> > **Rebuttal acknowlegement**
> >
> > Thank you for addressing my concerns. I appreciate the additional effort of running experiments on LLaMa 3 and doing the ablation study. I'll update my score.

---

> > > ### Author Response · Authors · 2024-06-06
> > >
> > > Thank you for considering our response and raising your score!

---

### Official Review · Reviewer_ShE4 · 2024-05-12

**Rating:** 7
**Confidence:** 4
**Ethics Flag:** 1

**Summary:**

This manuscript discusses an approach for improving fine-grained entity typing in a zero/few shot setting with LLMs, specifically GPT-4, by combining retrieval-augmented generation (RAG) with a knowledge base generated from a domain-specific ontology. The application area is chosen to be "highly specialized or fast-evolving themes," with the idea being that general models would not have been trained sufficiently in these areas. This is because the relevant distinctions would have appeared after the training cutoff for the fast-evolving themes, and because the relevant distinctions for highly specialized domain would be too specialized to appear in the training data.

The basic approach is to use the ontology to construct a knowledge base consisting of a "knowledge unit" for each node. The system then uses a RAG approach with a untuned LLM to perform the task, but in addition to returning the knowledge unit that best matches the query, the search component also returns the knowledge units for closely related nodes. The evaluation selects one highly specialized domain (redox-active organic electrode materials, a type of battery) and one fast-evolving domain (Genshin Impact, an online game). The battery domain uses the Battery Interface Ontology, which contains approximately 20,000 entity types. The online game does not have an ontology, so the authors work with "senior players" to create one, with 73 entity types, and annotate documents. The evaluation corpus for the battery domain consists of 20 "research papers in this field with a focus on batteries and electrodes." The corpus contains 100 annotated entities (mentions?) and 484 types, note this implies multiple types per mention. The evaluation corpus for the online game domain also contains 20 documents, in English and Chinese, with 200 entities (mentions?) and 73 types. Both evaluation datasets list one "hard case" for each document, essentially a mention that cannot be typed correctly without using knowledge from the ontology. The evaluation uses a straightforward prompt and considers GPT3.5 and GPT-4 with and without ORAG, demonstrating substantial improvements for precision, recall and f1-score in all cases. The discussion focuses on two case studies, one "unseen entity" (a nickname for a weapon in the online game that would only make sense in the context of the game) and one "misleading semantics" (a case where the specific discussion in context would seem to suggest a different, incorrect, type).

**Questions To Authors:**

Could you describe some concrete applications where this approach would be useful to users?

Please clarify the evaluation metrics

What are the error bars on the evaluation metrics from Table 2 & 3?

Please clarify the document-selection process, annotation guidelines, annotation process, inter-annotator agreement and ontology creation process.

Please clarify whether the full battery ontology was used during the evaluation or only the part relevant to the 484 types in the dataset.

How do you define "hallucination"?

How well would this approach scale?

How applicable would this approach be to other domains? How do you know?

**Reasons To Accept:**

This approach neatly addresses the need for domain-specific logic when using general LLMs in a specialized domain. The approach requires no modifications or fine-tuning for the LLM, and thus can be used in zero-shot settings for any LLM that already supports RAG. The approach is straightforward and seems relatively domain-independent, and thus should be widely implementable. The evaluation supports that the idea works, at least for the domains chosen. Overall, this seems like a viable way to add specialized semantics to an LLM without fine-tuning. It would also be orthogonal to approaches that would fine-tune the LLM, though this is not evaluated/tested in this manuscript.

**Reasons To Reject:**

There are several issues with the manuscript that reduce my enthusiasm for it.

First, the evaluation corpora are quite small: each is only 20 documents and the total number of entities (I think they mean mentions) is only 300. Thus, even with the substantial performance improvements described, these evaluations should not be considered definitive, even for the two domains described. The results for "hard cases" are even more uncertain, since they are based on only 20 mentions for each domain. I believe error bars on the evaluation (Tables 2 & 3) would help substantially, but even so, I think this work is best described as a pilot study - a successful one that clearly suggests the approach is worth large-scale evaluation, but still a pilot study.

Second, only two domains are considered. There are many others which this method should help, such as chemical, biological, and clinical research, which have substantial datasets that should be adaptable to FET. It is possible, however, that the authors do not consider these to be sufficiently specialized or fast moving. In any case, with only two domains considered, it is likely that there are substantial issues in some domains that would complicate its use or cause it to fail. Again, I think this work is best described as a successful pilot study.

It is also somewhat unclear in what applications this approach would be most useful. It requires a fairly comprehensive ontology, which must be created if it does not already exist, which implies a substantial effort. This effort itself would require substantial time and expense to evaluate examples from documents, which suggests that at least in the case of domains that do not already have an ontology, the approach should ideally be compared to creating a small training corpus and fine-tuning an LLM. The approach also requires one LLM prompt per document, mention pair, so likely cannot be applied to large-scale datasets with GPT-4: if we assume the prompt and response are 1000 tokens on average (750 words) and 100 mentions per document, then a dataset of only 10,000 documents would cost $45,000 (this analysis ignores the RAG component, which I assume would also cost tokens). Finally, fine-grained entity typing itself is probably best understood as an intermediate natural language processing task, one that would be useful in a pipeline but one that users are not likely to care about directly. The most straightforward application users might care about, building or expanding an existing knowledge base, would likely run into problems scaling. At the very least, it would be helpful to suggest some user-facing applications.

In addition to the evaluation being small, the manuscript provides only a very superficial description of the process for creating the evaluation datasets. Not described are: how the documents were selected, the annotation process (the number of annotators per document, how disagreements were handled, etc.), the annotation guidelines (what constitutes a mention and what does not), inter-annotator agreement, how the online game ontology was created (e.g. the number of entity types in the ontology is the same as the number in the corpus, suggesting that the ontology was created from the corpus), whether the knowledge units for the battery domain included all 20,000 entity types or only the 484 in the dataset. All of these are relevant for understanding the quality of the evaluation datasets.

The manuscript does not clarify how performance was measured. Is the performance for each mention (i.e., two mentions of "fruit platter" in a single document would count twice), or for each (document, mention text) pair, or something else? How are true positives calculated, given that each mention may be annotated with multiple entity types, each of which is hierarchical? The authors allude to a hierarchical evaluation in section 4 ("Stopping at correct but coarse grained...") but do not provide details.

Some may argue that the straightforward approach described is too incremental or obvious to be useful. While I know of related methods, I am not aware of any that combine ontologies with RAG so neatly and in a way that would be useful in a zero-shot setting. Thus, I specifically argue that any such argument against this manuscript should be accompanied by a citation of one or more works that allow an LLM to use RAG with an ontology in a zero-shot setting.

There are several clarifications or terminology changes that would be helpful:

Fine-grained entity typing (FET) is not as well-known as named entity recognition (NER) and entity linking (EL). A brief contrast between them would help the reader better understand the needs of the task. In this case, NER typically involves predicting span and rough entity type, while EL involves predicting a specific identifier from a controlled terminology or lexicon that is potentially very large (millions of entities). Fine-grained entity typing does not involve predicting the span, but does involve predicting a type from a set of types which are intermediate in size between the rough entity types of NER and the identifiers in EL. FET may also have a context dependency such that a particular aspect or role of the entity is emphasized, such as classifying a mention of a person that is a politician, business person and media personality as referring in context specifically to "person/politician"; in this sense FET may be more specific than even EL (see, e.g., Dai et al., EMNLP-IJCNLP 2019, Improving Fine-grained Entity Typing with Entity Linking https://aclanthology.org/D19-1643)

The authors use the word "entity" ambiguously, to refer to the type from the ontology and also to the text from the document that should be typed. The latter is more typically known as a "mention," and I strongly suggest updating this terminology since it makes the manuscript difficult to read and makes some of the descriptions unclear (e.g., doe the "entities" in Table 1 refer to mentions?)

The authors use the word "theme" where I believe that the field more typically uses the word "domain." This is not critical, since I don't think there is any ambiguity, but it would align the manuscript better with the field.

Finally, while the LLM literature uses the word "hallucination" rather liberally, this manuscript not only does not define it but uses it to describe almost any sort of error made by the LLM. This sort of imprecise language makes understanding the problems being addressed and analyzing the results more difficult. It also fails to make one of the strongest points in favor of this approach: Subbarao Kambhampati (e.g. https://twitter.com/rao2z/status/1772636256184520981) talks about LLMs being able to capture a distribution, but correctness is a property of an instance; by extension, "hallucination" is where the LLM returns something that sounds correct because it follows the distribution of the domain, but is not correct because the LLM has no knowledge of the entities within the domain. Thus, the approach described by the authors neatly allows the ontology to inform the LLM of the entities, which should dramatically reduce hallucinations, according to this definition, but the terminology used in the manuscript is not precise enough to support this point. (I'm also unsure whether the task would even support this sort of analysis.)

---

> ### Author Rebuttal · Authors · 2024-05-31
>
> Thanks for your insightful feedback and thoughtful queries. We aim to respond concisely within the length limit.
>
> **Themes vs. Domains**. Compared to domains (e.g., chemical and biological research), we use “themes” to refer to much narrower topics. Themes feature long-tailed, in-depth, or fast-evolving knowledge (e.g., “redox-active organic electrode materials” is a theme under the chemistry domain)
>
> **Error bars**. On the battery theme, for ORAG-GPT-3.5, SD(P) =0.11, SD(R) =0.28, SD(F1) =0.21. The paired t-test p-value over RAG is 0.008, showing our improvement is statistically significant. We omit other statistics to save space.
>
> **Experiment processes**. Each theme has two annotators. The primary annotator selects documents and performs the first round of annotation. Then, the predicted results from all models are pooled together and delivered to the primary annotator for a second round of annotation. After this, the secondary annotator checks the results and resolves disagreements with the primary annotator. The ontology is created and checked similarly.
>
> **Concrete applications**. The battery theme is inspired by a request from our industry partner to discover novel battery materials from text. The game theme originated from our observation that AI tools often make mistakes on game-related topics.
>
> **Evaluation metrics**. You are right that evaluation is done at the mention level. When multiple mentions in a document share the same surface name, we evaluate the first occurrence. The evaluation is multi-type, and parent types are auto-filled when a child type is predicted. Precision evaluates the percentage of predicted types that are ground-truth types. Recall evaluates the percentage of ground-truth types that are predicted.
>
> **Battery ontology**. We report the result using 484 relevant types instead of the full ontology.
>
> **Hallucination**. We use the word's broader meaning to refer to cases where LLM generates answers that look reasonable to ordinary people but are wrong based on expert judgment.
>
> **Scalability**. You may refer to the cost of running GPT-4. We add the experiments on the open-source model LLAMA3-8b-instruct and observe similar results. Please refer to our response to Reviewer YRhV for the performance table.
>
> **Generalizability**. Please refer to Points 1-3 of our response to Reviewer 28vN. There we also include results for an additional theme.
>
> We highly appreciate your detailed, constructive review and welcome further comments.

---

> > ### Comment · Reviewer_ShE4 · 2024-05-31
> >
> > Thank you for the response. Many of these make good sense, and I assume that there will be (minor) updates to the manuscript. I remain concerned about the size of the evaluation. I really do not believe it would be a good thing for the field to encourage additional work with such a small evaluation. And I am now concerned about the annotation process, which seems prone to bias. However, the ChemNER evaluation helps, and this work has my interest despite continued concerns. I also see that you adopted the "succcessful pilot study" phase, which I think does help frame the work. I will update my recommendation slightly.

---

> > > ### Author Response · Authors · 2024-05-31
> > >
> > > Thank you for your constructive feedback and for raising your rating!

---

### Official Review · Reviewer_28vN · 2024-05-17

**Rating:** 6
**Confidence:** 4
**Ethics Flag:** 1

**Summary:**

The paper proposes an Ontology guided RAG for the task of theme specific fine-grained entity typing. The paper argues that traditional datasets for entity typing, does not capture scenario with unseen entities or situations with misleading semantics, and constructs two datasets for evaluation under this scenario. Further, the paper proposes an ontology guided RAG algorithm, where the labels are enriched with expert knowledge and RAG is performed on top of these enriched ontology. Experimental evaluation shows ORAG leads to 10-24% improved F1 score as compared to traditional RAG.

The paper has good quality, clarity and originality. However, evaluation is done on a very small dataset of 20 documents, raising questions on the generalizability of the results.

Based on new results presented by the author in the rebutal phase, where they show generalizability of their approach to an existing CHEMNER dataset, I have updated my rating to "6: Marginally above acceptance threshold"

**Reasons To Accept:**

1. The paper argues that traditional datasets for entity typing, does not capture scenario with unseen entities or situations with misleading semantics, and constructs two datasets for evaluation under this scenario.
2. The paper proposes an ontology guided algorithm for RAG, including enriching the labels in the ontology. The proposed approach has novel contributions.
3. Empirical evaluation shows that the proposed ORAG approach has significantly improved performance over traditional RAG for the task of theme based entity typing.

**Reasons To Reject:**

1. Evaluation is done on a very small dataset of 20 documents, raising questions on the generalizability of the results.
2. Even the choice of themes - a) redox-active organic electrode materials, and b) online game posts for Genshin Impact, seems arbitrary without proper justification on how learnings on those datasets showcase generalizability.

---

> ### Author Rebuttal · Authors · 2024-05-31
>
> Thank you for recognizing our paper's quality, clarity, and originality. We understand your concern about generalizability, and we would like to respond to your concern as follows.
>
> 1. **Regarding the small dataset.** Our work focuses on the entity typing task on very narrow themes (e.g. redox-active organic electrode materials), which feature long-tailed knowledge with limited data. The number of documents per theme (20) is set to be small to align our experiments with such scenarios.
>
> 2. **Regarding the choice of themes and the generalizability.** The two themes are very different, demonstrating the **generalizability across themes**. The battery theme shows **generalizability in unseen types** because the types in the ontology are very hard to understand even for GPT-4 (e.g., the “Aluminum Insertion Electrode” type in Section 5.2). The game theme shows **generalizability in unseen entities** because the game is evolving very fast, and about half of the entities emerged very recently (e.g., four entities in Example Passage 2 are non-existent in the GPT pre-training corpus).
>
> 3. **To further demonstrate generalizability**, we add an additional experiment on a **chemistry theme “organic reactions”** using the ontology from a previous work called ChemNER. Results are shown in the following table. The conclusions are consistent with the two other themes in our submission.  If you have a particular theme of interest, you are welcome to let us know and we may try it out.
> | Method | Precision | Recall | F1 |
> | --- | --- | --- | --- |
> | RAG-GPT-4 | 0.89 | 0.64 | 0.75 |
> | ORAG-GPT-4 (Ours) | **0.91** | **0.83** | **0.87** |
>
> 4. Our methodology is designed to be **theme-independent** and generalizable. We would like to gratefully refer to Reviewer ShE4’s “Reasons To Accept” section for this point.
>
> 5. Despite all the above discussions, the scope of our experiments is still limited. Thus, we agree with Reviewer ShE4’s evaluation of our work as a **successful pilot study** that clearly suggests the approach is worth large-scale evaluation. Creating annotations on more themes is challenging as it requires expertise and can hardly be crowdsourced. We would greatly appreciate it if you kindly consider this pilot study to be accepted for publication to attract attention and resources for constructing large-scale theme-specific datasets.
>
> We welcome further comments and discussions.
>
> Update: Thank you for raising your rating!

---

### Decision · Program_Chairs · 2024-07-10

**Decision:**

Accept

**Comment:**

The paper introduces a novel approach, Ontology-guided Retrieval-Augmented Generation (ORAG), for theme-specific entity typing using large language models (LLMs). It addresses critical challenges in fine-grained entity typing, particularly with unseen entities and misleading semantics, demonstrating significant performance gains over traditional methods. The inclusion of two curated datasets and empirical validation on specific domains (e.g., battery materials, online gaming) highlights the effectiveness of the proposed approach. Since all reviewers have recommended accepting this paper, I recommend acceptance.